# Transversal Competencies in Operating Room Nurses: A Hierarchical Task Analysis

**DOI:** 10.3390/nursrep15060200

**Published:** 2025-06-03

**Authors:** Francesca Reato, Dhurata Ivziku, Marzia Lommi, Alessia Bresil, Anna Andreotti, Chiara D’Angelo, Mara Gorli, Mario Picozzi, Giulio Carcano

**Affiliations:** 1Department of Nursing Education, University of Insubria, 21100 Varese, Italy; francesca.reato@uninsubria.it; 2Department of Health Professions, Fondazione Policlinico Universitario Campus Bio-Medico, 00128 Rome, Italy; 3Department of Clinical and Molecular Medicine, Faculty of Medicine and Psychology, University LaSapienza, 00157 Rome, Italy; marzia.lommi@uniroma1.it; 4Operating Room Department, ASST dei Sette Laghi, 21100 Varese, Italy; alessia.bresil@asst-settelaghi.it; 5Operating Room Department, ASST Ovest Milanese, 20025 Legnano, Italy; anna.andreotti@asst-ovestmi.it; 6Department of Psychology, Università Cattolica del Sacro Cuore, 20123 Milano, Italy; chiara.dangelo@unicatt.it (C.D.); mara.gorli@unicatt.it (M.G.); 7Department of Biotechnologies and Life Sciences, Research Center in Clinical Ethics, University of Insubria, 21100 Varese, Italy; mario.picozzi@uninsubria.it; 8Department of Medicine and Innovation Technology (DiMIT), University of Insubria, 21100 Varese, Italy; giulio.carcano@uninsubria.it; 9Department of General, Emergency and Transplantation Surgery, ASST dei Sette Laghi, 21100 Varese, Italy

**Keywords:** soft skills, life skills, non-technical skills, transversal competencies, ethnography, hierarchical task analysis, HTA, nurses, operating room, perioperative, perianesthesiological

## Abstract

**Background:** Ensuring the safety of patients in the operating room, through the monitoring and prevention of adverse events is a central priority of healthcare delivery. In the professionalization of operating room nurses, the processes of identifying, assessing, developing, monitoring, and certifying transversal competencies are crucial. While national and international frameworks have attempted to define such competencies, they often vary in scope and remain inconsistently integrated into education and clinical practice. There is, therefore, a need for a comprehensive and structured identification of transversal competencies relevant to both perioperative and perianesthesiological nursing roles. **Objectives:** To formulate a validated and structured repertoire of transversal competencies demonstrated by operating room nurses in both perioperative and perianesthesiological contexts. **Methods:** A qualitative descriptive design was adopted, combining shadowed observation with Hierarchical Task Analysis (HTA). A convenience sample of 46 participants was recruited from a university and a public hospital in Italy. Data were collected between September 2021 and June 2023 and analyzed using content analysis and data triangulation. **Results:** Through a qualitative, inductive and iterative approach the study identified 15 transversal competencies, 50 sub-competencies, and 153 specific tasks and activities. Specifically, operating room nurses working in perioperative and perianesthesiological roles presented the following transversal competencies: communication and interpersonal relationships, situation awareness, teamwork, problem solving and decision-making, self-awareness, coping with stressors, resilience and fatigue management, leadership, coping with emotions, task and time management, ethical and sustainable thinking, adaptation to the context, critical thinking, learning through experiences, and data, information and digital content management. Each competency was associated with specific tasks observed. **Conclusions:** This framework complements the existing repertoire of technical-specialist competencies by integrating essential transversal competencies. It serves as a valuable tool for the assessment, validation, and certification of competencies related to patient and professional safety, emotional well-being, relational dynamics, and social competencies. The findings underscore the need for academic institutions to revise traditional training models and embed transversal competencies in both undergraduate and postgraduate nursing education.

## 1. Introduction

Operating rooms (ORs), along with intensive care units and emergency departments, are universally acknowledged as complex healthcare environments [1] suggestive to adverse events which constitute the second most prevalent type of medical error globally [2]. A considerable proportion of these adverse events are attributable to systemic failures encompassing issues related to outdated procedural workflows, or deficiencies in human factors such as inadequate training, miscommunication, and poor coordination [1]. While technical proficiency remains fundamental for the execution of clinical procedures, an equally critical determinant of safe and effective practice lies in the domain of non-technical skills. These skills include leadership, communication, teamwork, situational awareness, and decision-making [3].

To meet the evolving demands of healthcare complexity, it is imperative that both educational institutions and clinical organizations reconceptualize traditional models of training. This encompasses the enhancement of technical and scientific competencies and also the systematic integration of transversal competencies. This study employes the term “transversal competencies”, adopted from the European Skills, Competences, Qualifications and Occupations (ESCO) framework [4], to encompass the alternatively used terminologies like “non-technical skills” (NTS), “life skills”, “soft skills”, and “socio-emotional skills” [4]. Each of these terms refers to distinct yet overlapping domains of cognitive, interpersonal, and affective competencies that are not confined to disciplines or professions and are essential for adaptive performance in complex and dynamic work contexts.

Specifically, in healthcare, “non-technical” skills are defined as the cognitive and social abilities that underpin technical performance and contribute to patient safety and team efficiency [5]. The term “life skills” was first defined by the WHO [6] as “abilities for adaptive and positive behavior that enable individuals to deal effectively with the demands and challenges of everyday life”. “Soft” skills refer to personal and interpersonal traits that facilitate harmonious and effective interaction, including empathy, collaboration, and conflict resolution [7]. “Socio-emotional” skills, frequently emphasized within educational contexts, encompass the capacity to regulate one’s emotions, establish positive social relationships, and make responsible decisions [8,9]. The ESCO framework synthesizes these diverse terminologies into a unified classification of “transversal skills and competences (TSCs)” [4]. In this study, the “transversal skills” are defined as “learned and proven abilities which are necessary or valuable for effective action in any kind of work, learning, or life activity” [4].

Two major international frameworks have guided the integration of transversal competencies into educational and professional development pathways: UNICEF’s Global Framework on Transferable Skills [8] and the OECD’s Learning Compass 2030 [9]. UNICEF highlights skills such as empathy, negotiation, emotional regulation, and collaboration as essential for navigating personal, academic, and socio-economic domains [8]. The OECD, meanwhile, categorizes competencies into three overarching domains: cognitive and metacognitive skills (e.g., critical thinking, learning to learn), social and emotional skills (e.g., empathy, self-efficacy), and practical/physical skills [9]. Both frameworks underscore the universality and transferability of transversal competencies and their critical importance in preparing individuals for a rapidly changing world marked by technological, cultural, and organizational complexity [10].

Despite the growing recognition, the incorporation of transversal competencies into formal curricula and competency evaluation systems within healthcare remains inconsistent. In particular, there is an absence of systematic frameworks for measurement and certification of transversal competencies and their integration into nursing education and institutional performance metrics remains sporadic. However, a recent systematic review by Mullan et al. [11] identified and analyzed 31 observational instruments developed to assess transversal skills (TS) in the operating room including well noted taxonomies for TS in OR like NOTECHS, Oxford NOTECHS II, NOTSS, OTAS, SPLINTS, ANTS, N-ANTS, ANTS-AP. This resulted in the recognition of 47 TS competencies that span various healthcare professions. Nevertheless, the review revealed considerable heterogeneity in terminology, constructs, and application contexts, highlighting the urgent need for a unified, role-specific framework to guide education, assessment, and professional development in high-risk clinical settings.

Additional search of the literature regarding TS in OR allowed for the identification of other instruments like Operating Room Management Attitudes Questionnaire (ORMAQ) [12], the Modified SEIPS model (Systems Engineering Initiative for Patient Safety) [13] and the AS-NTS (Anaesthesiology Students’ Non-Technical Skills) [14]. In some studies, TS observations are structured around observable behaviors related to situational awareness, decision-making, communication and teamwork. These are valuable tools for performance evaluation, but they do not accurately reflect the full complexity of the perioperative and perianesthesicological environment. Some frameworks are highly task-oriented and emphasize team coordination during specific phases of surgical and anesthesiological procedures, while others attempt to address the specificities of nursing roles, yet tend to remain anchored in national or institutional contexts, lacking broader integration.

More recent contributions move toward a systems-based perspective, considering factors like organizational culture, work design, and latent conditions for error. However, these tools often overlook the individual professional’s capacity for adaptation, ethical reasoning, or reflexivity.

Each taxonomy captures partial aspects of a broader professional profile that is increasingly required in the operating room: one that demands not only cognitive and interpersonal skills, but also emotional resilience, systems thinking, ethical awareness, and the ability to learn and innovate continuously. The present study addresses this gap by proposing a comprehensive and integrative framework that brings together these dispersed elements.

Within high-complexity environments, operating room nurses hold a central role that transcends mere procedural execution. As key members of multidisciplinary surgical and anesthesiological teams, they must mobilize a broad repertoire of transversal competencies to ensure both patient safety and procedural efficiency. These competencies including communication, vigilance, team coordination, and rapid decision-making, are crucial for managing the unpredictable, time-sensitive scenarios that characterize perioperative and perianesthesiological care. However, traditional nursing education often fail to explicitly delineate or systematically cultivate these skills.

To our knowledge, no formal national framework currently exists in Italy that comprehensively maps transversal competencies for operating room nurses. This study seeks to fill this gap by employing Hierarchical Task Analysis (HTA), a methodological approach that utilizes a systematic decomposition of complex professional roles into their constituent tasks, responsibilities, and the competencies required to perform them [15]. The study will specifically focus in the development of a structured and validated repertoire of transversal competencies demonstrated and mobilized by operating room nurses in both perioperative and perianesthesiological roles. The findings from this research can contribute to the accreditation and certification of professional profiles, support development of evidence-based educational curricular reforms, and enhance workforce capacity building in high-stakes surgical environments.

Therefore, this study will explore the following research questions:(a)Which transversal competencies are demonstrated and mobilized by nurses in perioperative and perianesthesiological operating room contexts?(b)Which specific activities and tasks correspond to the application of these transversal competencies in both roles?

## 2. Materials and Methods

### 2.1. Design

This study used a qualitative descriptive design combining shadowing-based naturalistic observation with Hierarchical Task Analysis (HTA) to explore and map the real-world practices and transversal competencies of operating room nurses [16,17].

Shadowing is a real-time, immersive technique in which the researcher closely follows participants, observes them for a period of time during their routine professional activities, collecting data on observable behaviors and contextual nuances that influence performance, decision-making, and interaction patterns within natural settings [17,18]. In this study, shadowing was conducted in a non-intrusive manner, with the researcher observing nurses in perioperative and perianesthesiological roles. Field notes were recorded systematically in notebooks or post its, observing nurses’ actions, decision points, disruptions, team dynamics, and real-time adaptations to evolving clinical contexts. In addition to passive observation, the researcher posed clarifying questions to participants during or immediately after task execution. This technique, often referred to as a form of “continuous commentary”, enriched the data by uncovering the intentions and meanings behind actions and by capturing informal interactions otherwise overlooked in structured observations [17]. It also allowed for the analysis of both collective practices and individual decision-making processes, essential for a comprehensive understanding of professional conduct.

The observational data collected through shadowing were subsequently subjected to Hierarchical Task Analysis (HTA). HTA is a structured methodology used to deconstruct work processes into goals, sub-goals, and operational steps [16,18]. HTA enabled the systematic mapping of tasks and their underlying cognitive and behavioral components, facilitating the identification of transversal competencies that are often tacit and not easily accessible through self-report instruments. The integration of shadowing and HTA thus provided a robust framework for analyzing complex, context-dependent nursing activities in high-reliability settings such as the operating room.

### 2.2. Theoretical Framework

The development of transversal competencies of operating room nurses was constructed in light of the European Qualifications Framework (EQF) [19,20], the National Qualifications Framework (NQF) [21] and European Skills, Competences, Qualifications and Occupations framework (ESCO) [4]. The new Skills Agenda for Europe, proposed by the European Commission, as well as the review of the European Qualifications Framework and the European Key Competencies Framework for lifelong learning, led to the development of Competence Frameworks. Those taken into account in this study are: digital competencies (DigComp) [22], entrepreneurial competencies (EntreComp) [23], social, personal and learning competences (LifeComp) [24], and sustainability competencies (GreenComp) [25]. The European Qualifications Framework, the Italian Qualifications Framework and the European Competencies Frameworks have provided some useful inputs for the creation of the Operating Room Nurses Transversal Competencies Repertoire, both in terms of structure and content.

In the context of the EQF “competence” means the proven ability to use knowledge, skills and personal, social and/or methodological abilities, in work or study situations and in professional and personal development” [19,20]. This study focused solely on exploring the transversal competencies of operating room nurses in perioperative and perianesthesiological roles.

### 2.3. Study Setting and Participants

The study involved nurse managers, staff nurses and nursing students enrolled in an Operating Room Specialization Master’s program in Northern Italy. Observations were conducted between September 2021 and June 2023 using a convenience sampling strategy and voluntary participation.

To be included, the operating room nurses and nurse managers were requested to satisfy the criteria: (a) two years of tenure in the operating room setting; (b) work experience in perioperative and perianesthesiological nursing roles; and (c) agree to participate in the study without external solicitation. Nursing students of the Operating Room Specialization Master’s program were included if they: (a) actively enrolled in the Master during the data collection period, and (b) participated in the clinical internship in the operating room for a minimum of 500 h. The clinical placement experience was necessary to accurately recognize and interpret activities and tasks during participant observation.

Participants were excluded if: (a) not proficient in the Italian language; (b) work experience mainly in the sterilization unit; or (c) long absence from clinical practice (i.e., more than six months) or not actively working during the data collection phase.

All eligible individuals were informed about the study, and those who expressed interest and motivation independently contacted the research team. The final sample consisted of 6 nurse managers, 12 operating room nurses, and 28 nursing students of the Master’s program. A total of 22 nurses and 2 students were excluded because they did not satisfy the inclusion criteria. No participants withdrew or declined participation during the study data collection.

### 2.4. Data Collection

Data collection was conducted using shadowing, a qualitative observational method particularly suited to exploring complex, context-dependent professional activities in real-world settings. Shadowing involves the sustained, direct observation of nurses during their routine work, enabling the observer to follow their actions, interactions, and decision-making processes in real time and in situ [16,17]. A 4 h training on how to conduct shadowing observation (practical exercises in observation, field note-taking, and description of tasks) was conducted with all observers prior to data collection. Data collection was carried out by the Master’s program nursing students during their clinical internship activities.

Each observer shadowed nurses working in the perioperative and perianesthesiological roles in 5 shifts per month, over 6 months. Perioperative nursing encompasses all nursing interventions directly related to surgical procedures, including managing the operating field and instrumentation. Perianesthesiological nursing focuses on interventions related to anesthesia, covering the preparation room, operating room, and recovery room/post-anesthesia care unit. The aim of the observation was to comprehensively identify the transversal competencies and the maximum number of activities associated with the two operating room nursing roles under investigation. The data collection continued until data saturation was achieved.

The observer conducted continuous observations, closely accompanying OR nurses throughout their shifts without interrupting clinical activities. Observations were supplemented with brief, contextual inquiries during moments of reduced task demand, allowing clarification of task-related intentions, thought processes, and interpersonal dynamics. This interactive approach, often referred to as “conversational shadowing”, generated rich, real-time insights and a “continuous commentary” on the observed practices [16].

Detailed field notes were systematically recorded during and immediately after observations. The observers recorded each activity, described it with fundamental words and anchored it to transversal competencies on separate post-its. Observers were instructed to record as many distinct activities as possible, each on a separate post-it note, ensuring no repetition. The notes were brief to avoid disrupting the flow of observation but included essential details regarding who was doing what, when, and how [26]. This allowed observers to capture a comprehensive array of activities, contributing to a thorough description and facilitating the achievement of data saturation.

This method provided the empirical basis for the subsequent Hierarchical Task Analysis (HTA), by enabling the identification and categorization of discrete tasks, sub-tasks, and associated cognitive and behavioral demands.

### 2.5. Data Analysis

Inductive content analysis and data triangulation were employed for the analysis. The qualitative data collected through shadowing were analyzed using Hierarchical Task Analysis (HTA), a structured method used to deconstruct complex professional activities into their constituent goals, sub-goals, and operational steps [18]. HTA focuses on identifying the overall TCs and breaking them down into a hierarchy of sub-competencies and subordinate tasks necessary for their achievement. The analysis proceeded in multiple iterative stages [15]. First, the field notes which registered the observed sequences of actions, decision points, and task dependencies were reviewed to identify recurring verbal expressions and keywords as semantic units to code the tasks/activities (first level codes). Following, the words of the first level codes were grouped into specific and measurable conceptual clusters (sub-competencies) reflecting the practical skills needed in real-world. This grouping was based on a qualitative, inductive, and iterative approach. Next, the sub-competencies were grouped into broader transversal macro-competencies, based on semantic coherence, frequency of co-occurrence in the data, and professional relevance as observed during perioperative and perianesthesiological nursing care. A macro-competency uses a synthetic and overarching expression that integrates sub-competencies and associated tasks essential for effective professional performance across contexts. An example of how observed behaviors were categorized and abstracted in TCs is available in Appendix A.

Each task was defined according to its purpose, conditions for initiation and completion, and its relationship to broader competencies. Where applicable, plans (i.e., rules that govern the order and conditionality of task execution) were described using Annett’s notation system [18]. This hierarchical structure allowed for the integration of cognitive, behavioral, and interactional dimensions of work, capturing transversal competencies involved in real-world task execution.

To strengthen the validity and reliability of the analysis, a verification step involving subject-matter experts (SMEs) was incorporated. Following Annett’s guidance [18], experienced operating room nurses, nurse managers, observers and clinical educators with extensive domain expertise participated in 10 focus groups to draft and review HTA tables. Each group session lasted approximately two hours. During these sessions, the initial focus was on refining and addressing overlapping activities. Exact duplicates were eliminated, while semantically equivalent activities were consolidated through the combination of information using post-it notes. When a single activity had been attributed to multiple TCs, the group engaged in consensus-based discussion to determine the most contextually appropriate competency for classification. The selected task was then retained under the agreed-upon category and removed from others. This expert confrontation served two purposes: first, to confirm the completeness and accuracy of the task decomposition, and second, to promote ownership and acceptance of the analytical output by those most familiar with the professional context. Expert feedback was used to refine task definitions, clarify sequencing, and ensure that the HTA accurately reflected real-world nursing practice within the specific clinical setting. The final HTA tables were validated through a process of analytic triangulation, including cross-checking observational data with literature-based competency models, in order to ensure content validity and contextual accuracy. Indeed, the use of specific terms in the etiquettes of sub-competencies and competencies was not arbitrary: they were consistent with both the professional language observed in the field and the main competency taxonomies reported in the literature. The resulting hierarchical structure of transversal competencies reflects the functional and cognitive organization of the observed activities/tasks, structured into sub-competencies, and macro-competencies, in alignment with the methodological principles of HTA.

HTA was selected for its utility in high-reliability healthcare domains such as surgical procedures and emergency response systems [27,28]. Its use in this study enabled a detailed mapping of nursing practice and the identification of embedded transversal competencies that are central to effective performance in the operating room.

Throughout this process, the principles of rigor and reflexivity were applied in an integrated manner to ensure the validity and reliability of the findings [18]. Validation was ensured through participant discussions and regular team meetings. The data that emerged from the entire analysis process were synthesized and systematized within the Repertoire of Transversal Competencies.

### 2.6. Ethical Considerations

The study complies with the ethical standards and the principles of the Declaration of Helsinki [29]. The Board of Directors of the hospital and the University authorized the research. The participants voluntarily adhered to the study and, after adequate information, signed the informed consent. Data access was restricted solely to the research team.

## 3. Results

A total of 46 individuals participated in the study, comprising 6 nurse managers, 12 staff nurses, and 28 postgraduate nursing students enrolled in the Operating Room Specialization Master’s program. The majority of the sample was females with a mean age of 36 years and a tenure of 12 years. Additional descriptive demographic data for the sample are summarized in Table 1.

The data collection employed shadowing observation, during which observed behaviors were categorized into specific activities/tasks corresponding to distinct transversal competencies. Each participant completed 30 observation sessions within the operating room settings, yielding a cumulative total of 1380 observations. From the total dataset, 630 tasks pertaining to transversal competencies of OR nursing practice were identified and recorded on post-it notes.

The tasks were subjected to a rigorous process of systematization, which involved the integration, reduction, and synthesis of the identified activities/tasks, while maintaining consistency in etiquettes and aggregation with the established domains of competency. During ten meeting sessions with the expert team, the activities were discussed, duplicates removed, and similar tasks consolidated conceptually under unified descriptors of sub-competencies and transversal competencies. Through a process of functional and cognitive attribution, participants collaboratively mapped each task/activity to one or more relevant sub-competencies in operating room nursing, using a series of thematic boards, each dedicated to a specific competency domain. This operation contributed to greater clarity, internal consistency, and conciseness of the final repertoire of TCs. This process led to a reduction in the total number of tasks/activities included in the final list, without compromising the depth and representativeness of the content. On the contrary, the adoption of a shared and specific terminology strengthened the descriptive effectiveness of the entire repertoire. As a result of this process, 153 distinct tasks were retained and systematically organized across 50 sub-competencies and 15 defined transversal competencies. A summary of these outcomes is provided in Table 2. For the full list of tasks/activities associated with the competencies, refer to Appendix A.

Due to the extensive nature of the content, it is not feasible to present the entire TC table within this manuscript. Therefore, the authors have included an example of the HTA for transversal competency 7, as shown in Figure 1. The complete HTA for all competencies is available in Appendix A.

### Description of the Transversal Competencies Identified

The transversal competency “Communication and Interpersonal Relationship” is articulated through 12 tasks grouped in four sub-competencies: Communication, Conflict Management, Assertiveness, and Relationship. The results emphasize that effective communication is not limited to the transmission of information, but involves the creation of shared meaning through verbal, paraverbal, and non-verbal channels. Participants demonstrated the ability to actively listen and integrate feedback from surgical and anesthesiology teams, aligning expectations and maintaining clarity throughout procedures. Assertiveness emerged in the form of providing accurate, timely updates and ensuring mutual understanding through feedback mechanisms. Conflict management was reflected in the capacity to coordinate communication under pressure, especially in emergency contexts, maintaining calm and clarity. Finally, relational skills were evident in the use of tools to understand patient perspectives, ensuring personalized and compassionate care.

A real-life example of activities/tasks observed in clinical practice to support this competency is the following: “…*the unexperienced nurse chose dialog and communication with the surgeon and the anesthesiological experts to express her difficulties and doubts. She clarified her role and openly acknowledged her limited experience*…”.

The transversal competency “Situation Awareness” encompasses 11 tasks grouped in three sub-competencies: Situation Awareness, Focus, and Attention to Detail. The findings show that participants demonstrated a strong ability to anticipate potential complications, recognize evolving needs, and reduce response time, key aspects of maintaining situational vigilance in complex clinical environments. The Focus component involved preparing the body and mind through heightened alertness, enabling the perception of critical stimuli and the smooth transition of attention across multiple tasks. Under Attention to Detail, professionals showed advanced perceptual and cognitive control, recognizing environmental cues, prioritizing risks, and synthesizing complex information for informed decision-making. They actively monitored the surgical environment with a 360-degree perspective, paying close attention to signals, spatial layout, and workflow efficiency. The ability to manage resources, adapt to procedural changes, and project future developments based on real-time data further illustrates a robust capacity for dynamic assessment and intervention. An example regarding this competency can be found in this note: “…*the scrub nurse suggests maintaining a high level of attention, avoiding distractions and a careful observation…every action requires the timely and accurate preparation of the necessary instruments, without delays and manual dexterity… Attention must be paid to everything in the surrounding environment: avoiding contact with tables, carts, drapes, the entire set of instruments in use, and the hands of the operating surgeon*…”.

The transversal competency “Teamwork” comprises 10 tasks grouped in three sub-competencies: Teamwork, Coordination, and Interprofessional Collaboration. Participants demonstrated a consistent ability to respect roles and responsibilities, promoting inclusiveness and mutual support within the team. Coordination involved clear communication, responsiveness to colleagues’ needs, and smooth activity management. Interprofessional collaboration was particularly strong, with participants leveraging complementary skills, engaging dynamically, and fostering a cooperative culture. They worked across disciplines to ensure patient safety, supported one another in achieving shared goals, and applied strategic coordination to enhance performance and workflow efficiency. A clinical practice example is the following: “…*a scrub nurse emphasizes the fundamental importance of cooperation with the surgeon and the healthcare assistant…the open dialogue with the surgeon is one of the key elements that paves the way for effective collaboration. …listening to the surgeon’s proposals, respecting the surgeon’s preferences in the choice of instruments is both appropriate and necessary*…”.

The transversal competency “Problem Solving and Decision-Making” integrates 11 tasks in three sub-competencies: Problem Solving, Error Management, and Decision-Making. Participants exhibited the ability to clearly define problems, explore multiple creative solutions, and apply strategic reasoning to assess complex clinical scenarios. Error management practices were evident in the consistent application of safety protocols and proactive risk anticipation. Decision-making processes involved rapid information gathering, thorough evaluation of alternatives, and precise implementation. Accountability and reflective practice, both individual and collective, were central and decisions grounded in evidence and patient safety prioritized at all times. A practical example comes from this note: “…*one of the decisions she faces daily concerns how to set up the surgical field and the precise moment to proceed. In making these decisions, she relies on her knowledge of the time required to prepare all the necessary materials, the duration of hand scrubbing, gowning, and even the timing of anesthetic procedures. She highlights collaboration with the anesthesiologist and the anesthesia nurse as good practice, yet when the moment of decision arrives, she proceeds without hesitation…(a scrub nurse)*”.

The transversal competency “Self-Awareness” consists of 11 tasks grouped in two sub-competencies: Self Awareness and Self Efficacy. Participants demonstrated a clear understanding of their personal and professional limits, seeking support when necessary to ensure safe-care and continuous development. A strong commitment to self-reflection and self-care practices was evident. Under Self Efficacy, individuals consistently reflected on their goals, recognized their strengths and limitations, and approached challenges with resilience and confidence. They showed awareness of implicit biases, the impact of their behaviors on team dynamics and patient care, and emphasized respectful, patient-centered communication. Participants embraced constructive feedback, demonstrated clarity in role boundaries, and engaged in proactive, adaptive behaviors that strengthened both individual performance and collective team effectiveness. An example is given in the following: “…*she had little to no experience with the scheduled surgical procedures, she decided to request a replacement. To make this decision, she reflected on the few certainties she had, and she communicated her decision to the manager, explaining her reasoning and standing firm in her choice, even in the face of attempts at persuasion. She recommends gathering information about one’s level of preparedness before deciding and emphasizes the use of critical thinking and she underscores the importance of self-awareness…(a scrub nurse)*”.

The transversal competency “Coping with Stressor” is structured into 11 tasks grouped in three sub-competencies: Coping with Stressors, Coping Strategies, and Personal Well-Being. Participants showed a strong ability to remain calm and focused under pressure, even during prolonged or critical situations, demonstrating resilience and stress regulation. They adapted to high-stress environments by reflecting on actions, managing emotions, and making swift, clear decisions when challenges arose. Various proactive strategies were employed to manage stress, including relaxation techniques, planned breaks, and early identification of potential stressors. The emphasis on Personal Well-Being was evident in the use of self-care routines, emotional resilience practices, and seeking support when needed. Participants prioritized both mental and physical health, integrating restorative activities into their routines. An example is found on this note: “…*An anesthesia and recovery room nurse describes experiencing extremely high stress levels, primarily due to the intense attention required when administering a drug at a different concentration. …heightened attention (she refers to it as ‘obsessive attention’) is especially necessary during drug preparation…stress stems from the mental focus and repeated checks needed to avoid error… she explicitly reminds the anesthesiologist when a different concentration is being used*…”.

The transversal competency “Resilience and Fatigue Management” includes 11 tasks in four sub-competencies: Resilience, Reliability and Perseverance, Fatigue Management, and Workload Management. Participants showed the capacity to remain composed and determined in critical, high-pressure situations, effectively adapting to unforeseen changes with flexibility and a positive mindset. They maintained focus on goals, upheld commitments even during adversity, and demonstrated perseverance through reflective planning and goal adjustment. Fatigue was actively managed by recognizing its signs early, applying recovery strategies, and sustaining motivation in demanding contexts. The ability to protect both personal and team well-being, especially in prolonged or complex interventions—was evident, ensuring performance and productivity were not compromised. Here is an example to support this competency: “…*stress arises when, due to staff shortages, an anesthesia and recovery room nurse is required to manage two operating rooms simultaneously. The presence of an anesthesiology resident, with whom a productive collaboration and effective teamwork is established, helps to alleviate the perceived stress and workload. They agree on how to organize themselves and divide responsibilities between the two rooms at specific moments, highlighting abilities of collaboration, teamwork, coordinating and distributing tasks based on roles and responsibilities*…”.

The transversal competency “Leadership” is composed of 11 tasks grouped in four sub-competencies: Leadership, Ability to Delegate, Be Exemplary, and Taking Responsibility. Participants demonstrated a balanced leadership style, exercising authority with professionalism while avoiding authoritarianism. They maintained a results-oriented focus, managed resources responsibly, and adapted readily to dynamic clinical and team conditions. Effective delegation emerged through the fair distribution of tasks and active support for colleagues in difficulty. Leaders showed the ability to interpret situations strategically, plan interventions collaboratively, and incorporate diverse perspectives. They inspired team motivation, encouraged individual growth, and resolved conflicts with fairness. Leadership by example was particularly evident in crisis management, with individuals taking ownership of both personal and team actions and modeling shared values. The abovementioned description is another example of leadership abilities.

The transversal competency “Coping with Emotions” includes 9 tasks in four sub-competencies: Coping with Emotions, Emotional Management, Empathy, and Emotional Contagion. Participants demonstrated the ability to manage their own emotional responses in high-pressure situations, avoiding impulsivity and maintaining focus on patient safety. They effectively regulated emotions to prevent burnout, maintained clarity during moments of high emotional intensity, and contributed to a calm work environment. Emotional intelligence was promoted within teams through shared understanding, respectful communication, and empathetic engagement. Participants modeled emotional literacy and rational decision-making, encouraging a positive atmosphere where colleagues felt supported and valued. An example is the following: “…*There are differing opinions and choices that healthcare professionals may make when caring for women undergoing a voluntary termination of pregnancy (IVG): either to assist them or to exercise conscientious objection. In this case, the anesthesia and recovery room nurse chose to provide assistance and reports having felt empathy as soon as she met the patient. She recalls the gratitude expressed by the women she assisted…She refers to emotional detachment, a fundamental component of emotional regulation. It enables her to offer understanding while simultaneously protecting herself from being overwhelmed by others’ emotions to the point of being unable to function professionally*”.

The transversal competency “Task and Time Management” is composed of 9 tasks structured into four sub-competencies: Task Management, Time Management, Organization, and Anticipatory Thought. Participants displayed the ability to accurately plan surgical and anesthesiological activities, taking into account potential complications and resource needs. Task execution was based on clear prioritization and optimal use of time, materials, and equipment. Activities were planned with adherence to established protocols and standards, reflecting an evidence-based and structured organizational approach. Through anticipatory thinking, participants proactively identified risks, developed contingency plans, and addressed problems before they arose. They demonstrated initiative in streamlining procedural steps, ensuring efficiency, minimizing delays, and maintaining time productivity, especially during critical or urgent situations. An example of task management is presented in the following description: “…*once the patient has been anesthetized, she (the nurse) systematically plans all the necessary checks, proceeding from head to toe and establishing a set of priorities… the first essential check is to verify the patient’s posture and ensure that there are no risk situations, then proceeds to check everything at the level of the head (listing all the specific assessments she performs), next, she moves on to the chest (detailing the checks carried out), followed by the upper limbs, and continues in this manner, identifying priorities for each body area being assessed. She recommends developing a mental checklist to support the systematic execution of all required checks and to prevent any from being overlooked… (an anesthesia nurse)*”.

The transversal competency “Ethical and Sustainable Thinking” encompasses 12 tasks grouped in five sub-competencies: Sustainability, Ensuring Environmental Health, Ethical Awareness, Advocacy, and Legal, Ethical, and Deontological Accountability. Participants actively contributed to environmentally responsible practices by managing waste appropriately, promoting recycling and reuse, and selecting sustainable materials in the operating room. They demonstrated awareness of the ecological impact of surgical and anesthetic activities and helped implement sustainability programs aimed at reducing pollution and waste. Ethical awareness was reflected in patient-centered care practices aligned with personal and professional values. Advocacy emerged through respectful, inclusive communication and the safeguarding of patient dignity, privacy, and autonomy. Additionally, participants adhered to professional ethical codes, responded appropriately to unethical or unsafe practices, and supported a workplace culture grounded in integrity, accountability, and legal compliance. An example is the following: “(continuing from coping with emotions competency example) … *choices that healthcare professionals may make when caring for women undergoing a voluntary termination of pregnancy … In this case, the anesthesia and recovery room nurse offers a suggestion—given that this is an ethical dilemma in which the decision is highly personal and far from easy—to warn others that those in similar situations must be prepared to manage the many painful emotions experienced by the women involved. She highlights emotional management: not only regulating one’s own emotions, but also understanding and responding appropriately to the emotions of others*…”.

The transversal competency “Adaptation to the Context” is composed of 9 tasks on four sub-competencies: Cultural Adaptability, Adaptation to the Context, Adaptability, and Cultural Respect. Participants demonstrated the ability to collaborate with diverse teams by adjusting to different group dynamics and working respectfully with individuals from various backgrounds. They interacted with patients with cultural sensitivity, adapting communication styles and behaviors to meet individual preferences and expectations. Efforts to respect and accommodate cultural and linguistic diversity were evident in their ability to ensure team cohesion and patient-centered care. Participants also exhibited cultural respect by acknowledging beliefs, traditions, and social norms, and by tailoring care practices accordingly. Their actions supported the delivery of equitable, inclusive, and responsive care, reinforcing a healthcare environment that embraces diversity and promotes dignity and understanding for all. An example of this competency comes from the following: “…*In the operating room, the nurse should collaborate effectively with different teams, requiring adaptability to the specific relational and operational dynamics of each group, each characterized by its own culture, values, and modes of interaction. Attention and sensitivity are essential to respect and integrate these differences into daily practice*…”.

The transversal competency “Critical Thinking” is composed of 10 tasks grouped in two sub-competencies: Critical Thinking and Open and Critical Mindset. Participants demonstrated the ability to gather information objectively, distinguish between facts and opinions, and make decisions grounded in evidence. They consistently reflected on their actions and evaluated potential outcomes to ensure alignment with quality standards. With an open and analytical approach, individuals quickly assessed dynamic situations, formulated judgments based on solid reasoning, and posed relevant questions to resolve uncertainties. Acknowledging personal biases and limitations, they promoted critical dialog within teams to enhance decision-making. Intellectual curiosity and a willingness to reconsider views in light of new evidence supported a culture of continuous improvement and adaptability in clinical practice. An example of real-life activities is described here: “…*She (the scrub nurse) gathers information objectively, observes attentively and analyzes situations quickly, asks targeted questions to clarify uncertainties, reflects on every action, carefully evaluating the consequences to ensure quality and safety and bases decisions on concrete data. She fosters critical discussion and team reflection, encourages analytical thinking, and maintains an open mindset, ready to revise her views in light of new evidence*…”.

The transversal competency “Learning through Experience” includes 8 tasks grouped in three sub-competencies: Reflect, Learning to Learn, and Learning from Experiences. Participants demonstrated the ability to reflect on both successes and failures, using these insights to improve future goal-setting and performance. Feedback—both given and received—was leveraged as a tool for personal and collective growth. They actively sought opportunities for self-improvement, maintained updated clinical knowledge, and promoted a culture of continuous learning within the workplace. Intellectual curiosity was expressed through critical questioning and a proactive approach to expanding one’s knowledge. A strong commitment to lifelong learning was evident, with individuals engaging in ongoing education and development to remain adaptable and competent throughout their careers. An example is the following: “…*The scrub nurse explicitly states that she follows a study plan, to which she dedicates time every evening, prioritizing the surgical procedures she is less familiar with. She recommends making study a priority in order to acquire knowledge and avoid being unprepared. She suggests asking colleagues for their personal notebooks on surgical steps and then creating one of your own, noting that each person’s mental framework is different. She further recommends using this notebook as a study tool and keeping it in the operating room for consultation whenever needed. In this way, one has accessible resources to draw upon. For specific procedural steps, she advises asking the surgeons directly, referring to them as additional valuable resources*…”.

The transversal competency “Data, Information, and Digital Content Management” includes 8 tasks grouped in two sub-competencies: Research, Evaluate and Manage Digital Content and Manage Data, Information, and Digital Content. Participants demonstrated digital literacy by navigating, assessing, and evaluating online institutional platforms to locate and retrieve relevant clinical data efficiently. They planned and conducted structured digital searches, applied systematic strategies to manage content, and ensured the accuracy and reliability of information. The ability to interpret, filter, and assess digital resources was supported by the integration of technology into daily perioperative and perianesthesiological practice. Collaboration through digital tools further enabled smooth communication and coordination within healthcare teams, reinforcing digital competence as essential to modern clinical workflows. A description from clinical practice is the following: “…*The OR nurse is expected to adopt a proactive and systematic approach in the use of digital technologies, applying them effectively for managing data and content throughout perioperative and perianaesthesiological clinical practice… Additionally, she (the OR nurse) must be capable of adapting to digital advancements by integrating new technologies into daily work, while collaborating and sharing data and information efficiently with the team enhancing communication and coordination* … *Additionally, it is essential to demonstrate flexibility in using technologies or instruments different from those normally used, and in managing role changes or new responsibilities, adapting effectively to the constantly evolving conditions of the operating room*…”.

## 4. Discussion

The findings of this study challenge the traditional approach to nursing competencies by broadening the focus beyond technical expertise to encompass a more holistic and systemic understanding of professional skills. While existing literature has predominantly emphasized the importance of technical and procedural competencies in perioperative and perianesthesiological nursing, this study highlights the critical role of transversal competencies in shaping individual performance, team dynamics and patient safety. The identification of 15 transversal competencies underscores the necessity of integrating a human-centered approach into clinical practice and training programs, acknowledging that the quality of care is deeply influenced by cognitive, relational, and emotional competencies.

The Hierarchical Task Analysis (HTA) identified several tasks and key competencies essential for operating room nurses. The findings of this study are consistent with the four major European competency frameworks [4] as well as with the existing literature on TCs in the operating room [11,12,13,14]. This research provides empirical evidence supporting transversal competencies among OR nurses in perioperative and perianaesthesiological roles in Italy, and reinforcing findings from previous studies in the field. Moreover, it expands current knowledge by highlighting underexplored transversal competencies and introducing new competencies to existing OR frameworks. The present study expands knowledge on the following transversal competencies: Adaptation to the Context, Ethical and Sustainable Thinking, Learning Through Experience, and Data, Information, and Digital Content Management. These competencies are underrepresented in some frameworks and absent in most.

Adaptation to the Context emerged as a unique transversal competency not previously codified in traditional TC taxonomies. This competency refers to nurses’ ability to gather and interpret information related to environmental parameters, cultural values, and both formal and informal operating room norms, and tailor their actions accordingly. Adaptation to the context is translated in adjusting behaviors to cultural differences, interacting sensitively with diverse individuals, and flexibly managing workflows in high-turnover settings. Perioperative and perianaesthesiological nurses also demonstrate this skill by monitoring workflow conditions, understanding organizational dynamics, and aligning actions with both formal protocols and informal institutional norms. Adaptation to the context is fundamental within OR settings for effective collaboration with diverse teams. Moreover, the ability to adapt to different situations and contexts is crucial to ensure culturally sensitive, safe, and personalized care [30,31], and workflow efficiency in surgical settings [32,33,34,35,36].

Another innovative findings of this study regards the transversal competency of ethical and sustainable thinking. While international organizations such as AORN [37] and EORNA [30] acknowledge the ethical competency dimension, they do not explicitly include sustainability as a core domain. By integrating these two aspects, the present study offers a more inclusive and updated perspective on ethical practice within perioperative and perianaesthesiological care, positioning ethical and sustainable thinking as a central component of competent OR nursing practice. This competency combines the nurse’s responsibility to uphold ethical principles such as patient dignity, equity, and privacy with a commitment to environmental sustainability. Tasks such as responsible clinical waste management, patient advocacy, and addressing ethical dilemmas in the surgical context have rarely been included in traditional taxonomies of transversal skills. Participants in this study, similarly to previous research [38,39], emphasized the importance of adopting environmentally responsible practices such as waste reduction, recycling, and the use of reusable materials to reduce the ecological footprint of surgical and anesthetic procedures. These actions align closely with the GreenComp framework [25] and the European Federation of Nurses (EFN) emphasis on integrating ethics into everyday nursing care [31]. Alongside sustainability, ethical awareness was consistently viewed as fundamental to clinical decision-making in this study. Nurses are expected to uphold patient dignity and privacy, ensure transparency in decision-making, and maintain high levels of professional accountability. The operating theater environment is becoming increasingly complex due to ongoing scientific, technological, structural, and organizational changes. As hyper-specialization progresses and the demand for humanized care grows, nurses are increasingly confronted with uncertainty and ethical tensions. In such contexts, consultation with clinical ethics experts may become necessary. A positive ethical climate was associated with enhanced nursing performance and more effective management of ethical challenges in perioperative care [40]. Furthermore, the integration of evidence-based practice with patient and team values enables nurses to navigate complex ethical dilemmas—such as weighing potential harm against therapeutic benefit—thereby supporting sound, ethically informed clinical decisions which are essential in the OR settings [41].

Learning Through Experience is a transversal competency largely observed in OR clinical settings. Participants emphasized the value of reflecting on both successes and failures to foster professional development and improve care quality. Nurses engaged in structured debriefings, gave and received feedback, and participated in reflective learning to improve future performance. Indeed, the literature reports that practices such as debriefings, case reviews, and outcome analysis support evidence-based learning and help identify areas for improvement [42]. The knowledge sharing through meetings, briefings, and informal discussions was identified as a key mechanism for building collective expertise and sustaining a reflective, team-oriented learning environment and enhancing collaboration [35,43,44]. Additionally, engagement in continuing education—through courses, workshops, and professional updates—was recognized as essential to maintaining clinical competence. While elements of this competency may be implicitly included in broader TC constructs like Critical Thinking or Situation Awareness, most traditional taxonomies offer limited focus on reflective practice. In contrast, frameworks like LifeComp [24] emphasize lifelong learning, curiosity, and openness to feedback as essential personal and professional competencies. Furthermore, EORNA [30] and AORN [37] documents explicitly support situated mentoring and continuous education as cornerstones of nursing development. This study adds to those perspectives by showing how learning through experience directly influences not only individual growth but also team dynamics and patient outcomes, reinforcing the need to recognize and cultivate this competency within clinical training programs.

Data, Information, and Digital Content Management competency represents another original contribution of this study. Nurses demonstrated strong digital literacy skills, including safe and accurate documentation practices, as well as the ability to effectively navigate complex digital platforms to support clinical decision-making and communication within the perioperative environment. These behaviors, although not traditionally considered in standard transversal competency frameworks, are central to the DigComp framework [22], which emphasizes digital responsibility, communication, and information management as core competencies for contemporary healthcare professionals. Moreover, both the AORN [37] and EORNA [30] standards acknowledge the importance of digital competence for perioperative nurses, highlighting crucial elements such as data security, traceability of surgical materials, and electronic documentation as integral components of care quality and patient safety. However, these organization standards tend to address this competency in a more general or implicit way and do not explore it with the same level of detail or with the systematic integration proposed by this study. Specifically, in the AORN document [37] is emphasized the importance of using evidence-based guidelines to ensure safe and high-quality practice. Although the importance of electronic documentation and information management is mentioned, there is no in-depth discussion of the specific digital competencies required of perioperative and perianaesthesiological nurses. Similarly, the EORNA [30] document identifies five key domains of competencies, some of which may imply the use of digital tools; however there is no section exclusively dedicated to digital competencies or digital literacy. The findings of this study go beyond these general acknowledgments, revealing that nurses play an active and strategic role in digital governance, contributing to the updating and optimization of clinical protocols and continuously adapting to the rapid technological advances that characterize modern surgical and anesthesiological care. This competency is essential for maintaining high standards of quality and ensuring continuity of care in increasingly digitalized clinical settings. This suggests a promising area for further development and refinement in existing competency models.

In addition, the above mentioned transversal competencies, this study identified other competencies already present in existing TC taxonomies [11,12,13,14] and in perioperative and perianesthesiological frameworks [30,37]: Communication and Interpersonal Relationships, Situation Awareness, Teamwork, Problem Solving and Decision-Making, Leadership, Task and Time Management, and Critical Thinking. Each of these transversal competencies is important in the OR clinical practice. For example, communication was found to improve coordination, minimizing misunderstandings, and enhancing team responsiveness during high-stakes clinical procedures [45,46]. Strong team collaboration, grounded in mutual respect and shared decision-making, enhances both safety outcomes and workflow efficiency in surgical settings [32,33,34,35]. Situational awareness skills enable nurses to perceive, interpret, and project relevant clinical information in real time, which is particularly crucial in the dynamic and high-risk environment of the operating room [47]. Continuous developing and sustaining situational awareness in the operating room, enhances real-time decision-making and support high-performing surgical teams allowing for personal and professional growth [48,49].

In addition, the TCs Self-Awareness, Coping with Stressors, Resilience and Fatigue Management and Coping with Emotions, emerged significantly in this study and are considered fundamental competencies in the OR nursing care. Indeed, the literature evidences that finding creative solutions to unforeseen situations, such as managing complications during surgical procedures and anesthetic management [50,51] and the ability to lead during critical situations was identified as a key factor in the success of interventions [42,52]. The capacity to assess scenarios quickly and formulate informed judgments directly contributes to patient safety and the quality of intraoperative care [51]. The need to prioritize activities, anticipate procedural steps, and allocate time and resources effectively is a key competency in OR nursing to avoid delays and optimize patient outcomes [53,54]. Collectively, these competencies equip nurses to perform at a high level in fast-paced, high-stakes environments, ensuring coordinated, timely, and safe surgical care. Therefore, the suggestion is to enhance those competencies in the present TC frameworks and to give them equal attention to other competencies.

These findings underline how transversal competencies are not marginal additions but central pillars of perioperative nursing. The study also aligns with broader European efforts to define transferable, observable, and measurable competencies within healthcare professions [31]. By integrating frameworks such as EntreComp [23], DigComp [22] LifeComp [24], and GreenComp [25], the proposed model in this study expands the traditional boundaries of perioperative competence and offers a comprehensive, forward-looking vision.

This study represents a natural and methodological continuation of the previous work from the research team [55] on the Technical Professional Specialist Competencies for Operating Room Nurses. The findings of the present study prompt a broader reflection on the role of transversal competencies in nursing practice, particularly in highly complex settings such as the operating room, thus offering a holistic and integrated perspective on the nursing role within the operating room. As with the previous study, this research also highlights the value of a field-based observational approach, which has proven effective in identifying transversal skills that are often less visible but crucial for the quality of care, patient safety, and the well-being of the surgical team. By integrating and expanding upon the framework outlined in the previous study, this contribution offers a unified and professionally mature vision of the operating room nurse’s profile, grounded in field research and supported by a robust theoretical framework.

In line with the objectives of the Standards and Guidelines for Quality Assurance in the European Higher Education Area [56], this study emphasizes the importance of clearly defining and making transparent the expected competencies throughout the educational path, promoting a learning approach based on measurable outcomes that are transferable across Europe and consistent with the real-world challenges of clinical practice. In this sense, the proposed integrated repertoire of technical-specialist and transversal competencies contributes to strengthening the recognition of the operating room nurse as a strategic, autonomous, and highly qualified figure within the contemporary healthcare landscape.

In conclusion, this study calls for a redefinition of professional training and evaluation standards to include these emerging transversal domains. Recognizing, assessing, and fostering such competencies will enhance care quality, safety, and responsiveness to contemporary healthcare challenges. The presence of a framework provides greater understanding and awareness, enabling framing and policy development in the perioperative and perianesthesiological context.

### 4.1. Strengths and Limitations

One of the key innovations of this study lies in its methodological approach. and large sample of participants. By employing a qualitative descriptive design with the shadowing techniques of observation and hierarchical task analysis, this research moves beyond self-reported competencies and explores how skills manifest in real-world clinical practice. This methodological choice allows for a deeper understanding of how competencies are enacted in complex and dynamic settings, where decision-making, situational awareness, and adaptability play a fundamental role in patient safety.

However, the study has several limitations. The finds come from a localized sample of operating room nurses from a single region and hospital. This could limit the generalizability of findings to broader international or multidisciplinary settings. The competencies were identified through shadowing observations and expert team discussions, which may introduce bias due to participants’ self-perception, social desirability, or recall limitations. Additionally, while the study offers deep insights into perioperative and perianesthesiological settings, its competency framework may not be fully applicable to nurses in other operating room settings depending on institutional policies, procedures or technological development. Furthermore, the study captured a snapshot of competencies at one point in time. It does not assess how these competencies develop or are sustained over time, particularly with changes in practice or ongoing education.

### 4.2. Implications for Practice and Education

The findings of this study have important implications for both nursing education and clinical practice. Transversal competencies, such as communication, teamwork, emotional intelligence, and ethical thinking are essential to safe and effective care. Yet, they are often underemphasized in traditional nursing curricula, which tend to mostly focus on technical skills.

Participants in this study consistently emphasized the importance of recognizing personal limits, engaging in self-reflection, and practicing self-care to sustain professional growth and prevent burnout. The promotion of emotional intelligence—through emotional literacy, recognition of affective states, and modulation of responses—was shown to enhance communication, reinforce team cohesion, and support a psychologically safe clinical environment [57,58,59]. An example are strategies such as mindfulness and breathing exercises that were reported as effective tools for maintaining composure and concentration during high-pressure situations [60,61].

Educational programs should embed transversal competencies using methods such as experiential training models, high- and low-fidelity simulations, reflective practice with guided feedback, and interdisciplinary learning [62]. Specific activities need to include the analysis of real clinical episodes to promote self-awareness and emotional recognition exercises.

The formal assessment and certification of transversal and technical-professional competencies could help shift professional standards toward a more holistic view of nursing expertise, valuing not only procedural skill but also collaboration, adaptability, and emotional regulation.

Longitudinal and interdisciplinary research is needed to evaluate how transversal competencies influence clinical performance and patient safety over time. Future studies should also focus on the development of standardized tools to assess these competencies and explore how institutional and organizational contexts shape their application. Fostering these skills is critical to developing resilient, adaptive, and human-centered nursing practices.

## 5. Conclusions

Through the application of Hierarchical Task Analysis, this study identified 15 transversal competencies and 153 related tasks and activities essential to the professional role of operating room nurses in both perioperative and perianesthesiological settings. These competencies encompassing areas such as communication, leadership, emotional intelligence, situational awareness, and ethical thinking. The transversal competencies identified in this study expand the set of transversal competencies present in the European Competency Domains and the previous research for the non-technical skills in operating room professionals.

The integration of transversal competencies in clinical practice is critical for ensuring patient safety, team performance, and high-quality care in complex surgical environments. Furthermore, the transversal competencies identified support the development of reflective practitioners who continuously adapt and grow in response to evolving healthcare challenges and ethical standards.

The framework serves as a foundation for competency-based education, certification, and professional development. It supports the integration of transversal competencies into nursing curricula and clinical training pathways. By formally recognizing and cultivating these skills, academic institutions and healthcare organizations can enhance the preparedness, adaptability, and resilience of surgical nursing teams. Standardized assessment and certification would further promote recognition of specialized nursing roles across healthcare systems. The existing debate within and among nations regarding the diversity of nursing education highlights the challenge of establishing common standards and levels of qualification. In this context, the proposed model offers a valuable contribution toward advancing comparability and improving quality standards across European and international nursing education systems.

Future research should explore the implementation of this framework in diverse clinical and educational contexts, and assess its impact on patient outcomes, interprofessional collaboration, and workforce sustainability. Longitudinal studies could further investigate how transversal competencies evolve over time and how they interact with organizational factors to shape professional identity and clinical performance.

## Figures and Tables

**Figure 1 nursrep-15-00200-f001:**
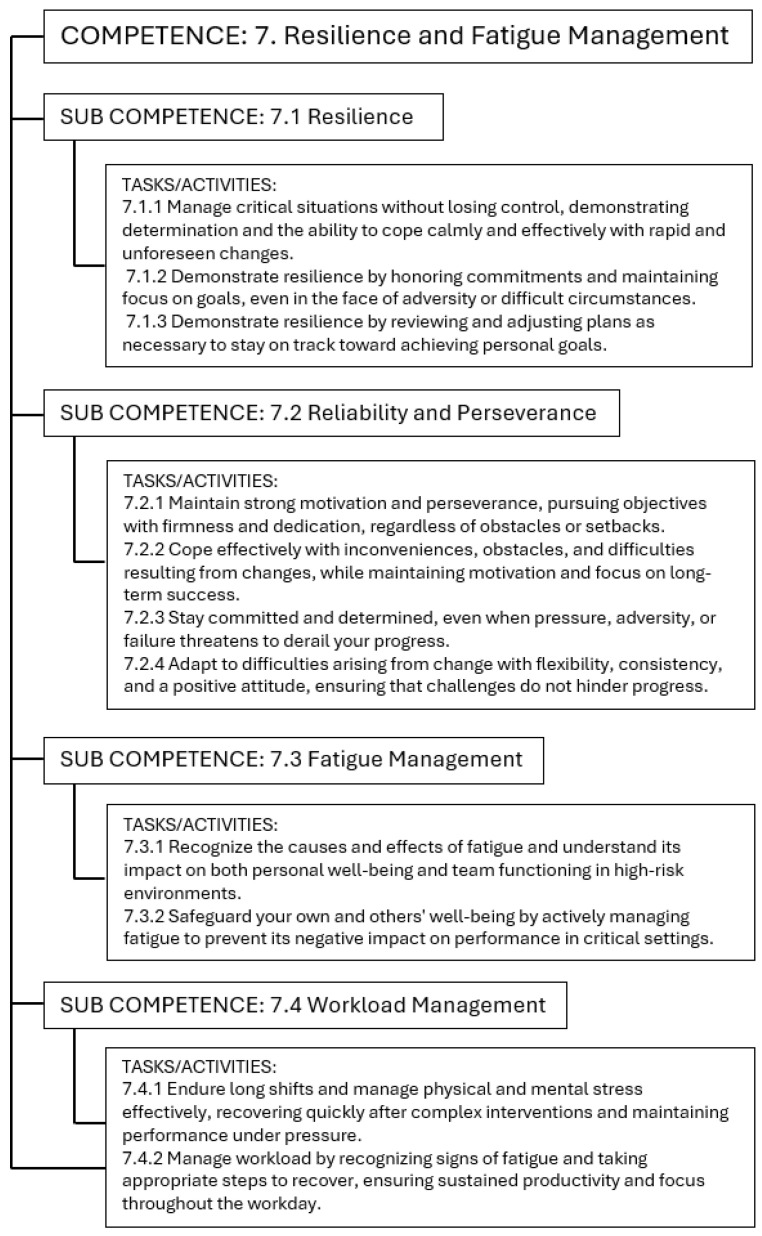
Hierarchical Task analysis presentation for transversal competency 7.

**Table 1 nursrep-15-00200-t001:** Sociodemographic sample characteristics (N = 46).

	N (%)
Sex	
Male	4 (8.7)
Female	42 (91.3)
Age (mean (SD)	36 (12.4)
Tenure in operating room (mean (SD)	12.68 (13.5) range (0–25)
Higher level of education	
Bachelor Degree	35 (76.1)
Specialization Master	10 (21.7)
Master’s Degree	-
Second level Specialization Master	1 (2.2)
Participating as	
Master’s student nurses	28 (60.8)
Nurses	12 (26.1)
Nurse managers	6 (13.1)

**Table 2 nursrep-15-00200-t002:** Summary of competencies, sub competencies and task with relative frequency.

N	Competencies (N = 15)	Sub-Competencies (N = 50)	Tasks Frequency (N = 153)
1	Communication and interpersonal relationships	Communication	12
		Conflict management	
		Assertiveness	
		Relationship	
2	Situation awareness	Situation awareness	11
		Focus	
		Attention to detail	
3	Teamwork	Teamwork	10
		Coordination	
		Interprofessional collaboration	
4	Problem Solving and Decision-Making	Problem Solving	11
		Error management	
		Decision-Making	
5	Self Awareness	Self Awareness	11
		Self Efficacy	
6	Coping with Stressor	Coping with Stressors	11
		Coping Strategies	
		Personal well-being	
7	Resilience and Fatigue Management	Resilience	11
		Reliability and Perseverance	
		Fatigue Management	
		Workload Management	
8	Leadership	Leadership	11
		Ability to delegate	
		Be exemplary	
		Taking Responsibility	
9	Coping with Emotions	Coping with emotions	9
		Emotional management	
		Empathy	
		Emotional contagion	
10	Task and Time Management	Task Management	9
		Time Management	
		Organization	
		Anticipatory thought	
11	Ethical and sustainable thinking	Sustainability	12
		Ensuring environmental health	
		Ethical awareness	
		Advocacy	
		Legal, ethical and deontological accountability	
12	Adaptation to the context	Cultural adaptability	9
		Adaptation to the context	
		Adaptability	
		Cultural Respect	
13	Critical Thinking	Critical Thinking	10
		Open and critical mindset	
14	Learning through experiences	Reflect	8
		Learning to learn	
		Learning from experiences	
15	Data, Information and Digital Content Management	Research, Evaluate and Manage Digital Content	8
		Manage data, information, and digital content	

## Data Availability

The data presented in this study are available on request from the corresponding author due to privacy restrictions.

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
