# Peer review of "Transversal Competencies in Operating Room Nurses: A Hierarchical Task Analysis"

_nursrep, 2025, doi:10.3390/nursrep15060200_

Round 1

Reviewer 1 Report

Comments and Suggestions for Authors

Title and abstract

-First of all, I congratulate the author/authors for their interest in this topic. For the title

  1. "Transversal competencies" may not be a very common term. Could more common terms such as "Cross-functional Competencies" or "Multidimensional Competencies" be preferred instead?
  2. "Hierarchical Task Analysis" is a very broad concept. Can it be stated more clearly which tasks the operating theater nurses will be focused on? For example, "Clinical Decision-Making Tasks" or "Communication and Coordination Tasks."

-The background section of the summary makes a very good start. However:

  1. The first two sentences are too general and feel a bit disorganized. These sentences could be combined and made more fluent.
  2. The term ‘Human Factors’ is used, but what exactly is meant here? It can be made more specific with subheadings such as training, communication, and decision-making.

Introduction:

-The introduction is quite comprehensive and very well structured. However, the paragraphs are long and dense. For this reason, some sentences could be shortened or divided into several paragraphs.

-Wording could be simplified where the same ideas are repeated. For example, concepts such as "non-technical skills" and "transversal competencies" are repeated, but a clear distinction could be made between them.

-The objective and contribution of the study could be emphasized more clearly.

Materials And Methods:

-Terms such as "shadowing" and "hierarchical task analysis" are repeated several times. Emphasizing the differences between these terms or adding a clearer definition for each of them could make the text more understandable.

-In the study settings section, the selection of participants and the scope of the study are explained in a very clear and detailed manner. The clear inclusion and exclusion criteria reinforce the validity and reliability of the study methodology. In addition, detailed explanations about the adequacy of the participants' professional experience and training processes show that great care has been taken to increase the validity of the data obtained during the observations.

Results

The findings obtained in this study provide extremely valuable and meaningful data for the purpose of the research.

Discussion

- In the discussion section, the transitions could be made a little clearer. For example: ‘Effective communication and interpersonal relationships are not only necessary for conveying accurate and timely information but also for fostering mutual understanding...’ to ‘Effective communication and interpersonal relationships are not only essential for the accurate and timely transmission of information, but also play a critical role in building mutual understanding...’ This makes the flow of the sentence smoother and emphasizes the connection between the different transversal competences.

- The emotional intelligence part is very relevant, but it may be useful to add more details about its applicability.

- The integration of transversal competences into training programs is very important, but specific suggestions on how to do this could be added.

Reviewer 2 Report

Comments and Suggestions for Authors

Dear Authors,

This study provides a comprehensive framework of transversal competencies demonstrated by operating room (OR) nurses in perioperative and perianesthesia contexts. The use of Hierarchical Task Analysis (HTA) is methodologically appropriate, and the identified set of competencies offers valuable implications for practice, education, and policy. The breadth of the analysis, including 15 transversal competencies and 153 associated tasks, represents a significant contribution.

However, the manuscript may benefit from structural refinements to enhance clarity, conceptual coherence, and communicative impact. Specifically, the connection between the stated research aims and the conclusions, the transparency of data transformation, and the interpretative depth of the discussion could be strengthened.

Please refer to the attached file for details. I hope that my comments will contribute, even in a small way, to the improvement of your manuscript.

Best regards,

Round 2

Reviewer 2 Report

Comments and Suggestions for Authors

Dear Authors,

Thank you for revising and adding content based on the reviewer comments within such a short period of time. I have confirmed that the comments have been appropriately addressed and reflected in the manuscript.

Best regards,